# Influence of PM_2.5_ Exposure Level on the Association between Alzheimer’s Disease and Allergic Rhinitis: A National Population-Based Cohort Study

**DOI:** 10.3390/ijerph16183357

**Published:** 2019-09-11

**Authors:** Ruo-Ling Li, Yung-Chyuan Ho, Ci-Wen Luo, Shiuan-Shinn Lee, Yu-Hsiang Kuan

**Affiliations:** 1Department of Public Health, Institute of Public Health, Chung Shan Medical University, Taichung 40201, Taiwan; 2Department of Medical Management, Division of Thoracic Surgery, Chung Shan Medical University Hospital, Taichung 40201, Taiwan; 3School of Medical Applied Chemistry, Chung Shan Medical University, Taichung 40201, Taiwan; 4Department of Pharmacology, School of Medicine, Chung Shan Medical University, Taichung 40201, Taiwan; 5Department of Pharmacy, Chung Shan Medical University Hospital, Taichung 40201, Taiwan

**Keywords:** Alzheimer’s disease, allergic rhinitis, particulate matter, PM_2.5_, cohort study, Kaplan–Meier curves

## Abstract

Alzheimer’s disease (AD) is an irreversible neurodegenerative disease that leads to dementia, health impairment, and high economic cost. Allergic rhinitis (AR) is a chronic inflammatory and allergic disease of the respiratory system that leads to health problems and has major effects on the daily lives of patients and their caregivers. Particulate matter (PM) refers to air pollutants 2.5 μm or less in diameter that are a source of concern because of their role in numerous diseases, including AR and other neurodegenerative diseases. To date, no study has demonstrated how PM_2.5_ exacerbates AR and results in AD. We conducted a national population-based cohort study by obtaining patient data from the National Health Insurance Research Database of Taiwan for the 2008–2013 period. PM_2.5_ concentration data were obtained from the ambient air quality monitoring network established by the Environmental Protection Administration of Taiwan. Monthly PM_2.5_ exposure levels were categorized into quartiles from Q1–Q4. The Cox proportional hazards analysis, after adjusting for age, sex, low income, and urbanization level, revealed that patients with AR had an elevated risk of developing AD (hazard ratio (HR): 2.008). In addition, the cumulative incidence of AD in the AR group was significantly higher than in the comparison group. The PM_2.5_ levels at Q2–Q4 (crude HR: 1.663–8.315; adjusted HR: 1.812–8.981) were stratified on the basis of the PM_2.5_ exposure group and revealed that AR patients exposed to PM_2.5_ are significantly prone to develop AD. In addition, the logistic regression analyses, after adjustment, demonstrated that an increase in the PM_2.5_ exposure level at Q2–Q4 (adjusted odds ratio (OR): 2.656–5.604) increased the risk of AR in AD patients. In conclusion, an increased PM_2.5_ exposure level could be correlated with AR, which could in turn cause AD. AR increased the risk of AD, in which exposure to PM_2.5_ increases the higher probability of AD.

## 1. Introduction

Alzheimer’s disease (AD) is an irreversible and progressive neurodegenerative and neuroinflammatory disease that leads to the impairment of memory, cognition, and behavior [1,2]. The major symptoms of AD are dementia and memory loss. Globally, 46.8 million people are living with AD or other forms of dementia, and this number is increasing rapidly and could rise to 152 million people in 2050 [1,3,4]. Today, someone develops Alzheimer’s dementia every 66 s and the number of individuals with Alzheimer’s dementia in the United States (U.S.) is approximately 5.5 million. By 2050, one new patient with Alzheimer’s dementia will receive a diagnosis every 33 s and 13.8 million Americans will live with Alzheimer’s dementia [5]. AD not only affects the lives of the patients but also those of their families and caretakers, with great economic cost [6]. The estimated total global cost of AD was higher than US $230 billion in 2016, including 18.2 billion care hours by 5 million family members and other unpaid caregivers [5].

Allergic rhinitis (AR), an IgE-mediated inflammatory and allergic disease of the respiratory system, is characterized by symptoms such as nasal congestion, anterior nasal leakage, airflow obstruction, posterior rhinorrhea, olfactory dysfunction, and sneezing [7]. Although AR may not cause serious health problems, the multiple symptoms associated with AR can have a considerable effect on the daily lives and physical, psychological, and social functioning of those affected [7]. In the U.S., the annual expenditure on public health is as high as US$0.2–5 billion, and the loss of revenue caused by loss in productivity is estimated to range from US$0.2 to 4 billion annually [8].

Epidemiological studies have revealed a rapid increase in the prevalence of AR in the past decades [9]. The incidence of AR globally has increased from less than 1% in the 1920s to 10–40% today [10,11]. Ambient air pollution is a major risk factor for AR [11]. Particle pollution, also called particulate matter or PM, exists in the atmosphere. PM_2.5_, referring to air pollutants or particulate matter with a size of 2.5 μm in diameter or less, is a concern because of its potential negative health effects. Indeed, PM_2.5_ is highly correlated with AR and influences its occurrence in China and Germany [12,13]. In addition, PM_2.5_ exposure is a major risk factor for neurological disorders, such as AD, stroke, dementia, autism spectrum disorder, and Parkinson’s disease [14]. Inflammatory response induced by allergy could be the important risk factor in the development of AD through tau phosphorylation [15]. To date, no study has explored whether AR is relevant to the risk of AD and the relationship between exposures to PM_2.5_. The present study investigated the relationship between AR and AD at various levels of PM_2.5_ exposure.

## 2. Materials and Methods

### 2.1. Data Sources

In the present study, the patient data were sourced from the Longitudinal Health Insurance Database (LHID) of the National Health Insurance Database (NHIRD), which is the compulsory insurance database used primarily for reimbursement purposes in Taiwan. The NHIRD provides de-recognized files that contain information concerning healthcare services rendered to more than 99% of Taiwan’s 23 million residents. Data regarding inpatient and outpatient visits, admissions, and discharges in the NHIRD records are coded by physicians according to the International Classification of Disease, 9th Revision, Clinical Modification (ICD-9).

### 2.2. Collections and Concentrations of PM_2.5_

The Environmental Protection Administration of Taiwan (TEPA) set up the ambient air quality monitoring network (AQMN) in Taiwan to monitor general air quality [16]. The PM_2.5_ concentrations in the atmosphere were measured using a tapered element oscillating microbalance (R&P 1400, Rupprecht and Patashnick, Inc., New York, NY, , US). The measured values were recorded every hour. The protocol of study was approved by the Ethics Review Committee of Chung Shan Medical University Hospital (CS215106). Since the time unit used during follow-up period was the month, this study used the monthly average cumulative exposure as a comparison of patient exposure. The patient exposure was based on the exposure of the station where the patient is located. If there is no station at the location of the patient, it was based on the nearest station. The monthly average cumulative exposure of the patient was observed from 2008 to the end point. The cumulative exposure per hour was multiplied by 24 as the basis for daily exposure, which was excluded if there were more than 8 h of observed omissions. The average daily exposure is multiplied by the number of days in the month is the monthly average exposure, which is excluded if there are more than 10 days of observed omissions. The monthly average PM_2.5_ concentration was estimated and considered as the population exposure at the county level using data from AQMN in Taiwan. We refer to the study using quartiles as the basis for grouping exposure concentrations [17] To corroborate the exposure–response relationship, the monthly concentrations of PM_2.5_ were categorized into the following quartiles: <25th percentile (Q1), <646 μg/mm^3^; 25–50th percentile (Q2) 646–1253 μg/mm^3^; 50–75th percentile (Q3), 1253–2056 μg/mm^3^; and >75th percentile (Q4), >2056 μg/mm^3^.

### 2.3. Study Population

Datasets for the years 2008–2013 from the LHID and AQMN were obtained for the present study. As illustrated in Figure 1, 429,277 individuals aged 30 or older who did not receive a diagnosis of AR (ICD9: 477, 477.0, 477.1, 477.8, 477.9) and AD (ICD-9: 331.0) before 2008 were recruited from the NHIRD. In addition, we excluded patients with incomplete basic data, including conflicting sex information, uncertain birth date, and living region with PM_2.5_ undetected from AQMN. From 2008–2013, 3803 patients were diagnosed with AD. All enrolled cases were divided into AR and non-AR (comparison) groups. In the AR group, the date of the first AR diagnosis was the index date, and then the patients visited doctors for AR treatment at least two times. In the comparison group, the index date was 1 January 2008. The date of AD diagnosis was considered the end point. We excluded patients with an end point before the index date.

The basic characteristics of the subjects included sex, age, low income, and urbanization levels. The age divided into young group (aged 30–44 years), middle-aged group (aged 45–59 years), young-old group (aged 60–74 years), and middle-old group (aged 75-years). Exemption of income tax represents family monthly income less than 20,000 NTD (New Taiwan Dollar) per month in Taiwan [18] (Lee et al., 2008). All 359 communities in Taiwan were stratified into the seven urbanization levels based on composite score including the population density, the proportion of educational level, the proportion of over aged 65 years, the proportion engaged as agricultural workers, and the number of physicians per 100,000 people. Seven urbanization levels include highly urbanized, moderately urbanized, emerging town, general town, aged township, agricultural town, remote township [19]. Comorbidities included hypertension (ICD-9: 401–405, 437.2), hyperlipidemia (ICD-9: 272, A189), hyperglycemia (ICD-9: 250, A181), Huntington’s disease (ICD-9: 333.4), Parkinson’s disease (ICD-9: 332), extrapyramidal signs (ICD-9: 333), sleep apnea (ICD-9: 780.51, 780.53, 780.57), and nutrition deficiencies (ICD-9: 260–269).

### 2.4. Statistical Analysis

The χ^2^ test was used for categorical variables during the analysis of the differences between the case and control groups. A two-tailed *t*-test was used to compare continuous variables. Multivariate stratified Cox regression models were then subsequently used to calculate the hazard ratio (HR) and 95% CI. Multivariable models were adjusted for AD-related co-morbidities, gender, age, low-income, and urbanization level. Kaplan–Meier Curves estimating cumulative incidence of AD between AR and non-AR patients. Conditional logistic regression was used to estimate the patients with AD odds ratios (ORs) and 95% confidence intervals for the risk factors of AR. Potential risk factors were sex, age, low income, other operations, and comorbidities. Statistical analyses were performed using SAS 9.3 software (SAS Institute, Cary, NC, USA), and *p* < 0.05 was considered statistically significant.

## 3. Results

### 3.1. Patient Characteristics

Table 1 presents the basic characteristics of AD patients with or without AR. In total, 2384 patients with AD aged 30 or older were selected from a nationally representative sample from Taiwan’s LHID in the 2008–2013 period. Among the patients with AD, 370 and 2014 were AR and comparison patients, respectively. There were no significant differences in sex, low income, age, and urbanization level between the AR and the comparison patients with AD. In addition, we observed a higher proportion of persons whose characteristics, such as age >75 (comparison group, 45.58%; AR group, 39.26%), low income (comparison group, 61.27%; AR group, 63.50%), and moderate urbanization (comparison group, 27.06%; AR group, 29.45%) in the patients with AD.

### 3.2. Kaplan–Meier Curves and Cox Regression

During the follow-up period, the cumulative incidence of AD in the AR group was significantly higher than that in the comparison group (*p* < 0.0001, Figure 2). The results indicated that patients with AR had a higher risk for AD. Similar findings were obtained from the Cox regression analysis, which revealed that patients with AR were at higher risk for AD (adjusted HR: 2.008, 95% CI: 1.780–2.266, *p* < 0.001; Table 2). Female (adjusted HR: 0.900, 95% CI: 0.829–0.978, *p* < 0.05) and low income (adjusted HR: 1.129, 95% CI: 1.028–1.240, *p* < 0.05) patients were the risk factor for AD. Patients lived in remote township (adjusted HR: 1.268, 95% CI: 1.026–1.567, *p* < 0.05) was a risk factor for AD. However, patients with comorbidities such as hypertension (adjusted HR: 0.771, 95% CI: 0.703–0.845, *p* <  0.001), hyperlipidemia (adjusted HR: 0.752, 95% CI: 0.703–0.845, *p* < 0.001), and extrapyramidal signs (adjusted HR: 0.773, 95% CI: 0.703–0.845, *p* < 0.001) were at a reduced risk compared with those with none.

Table 3 presents the results of Cox proportional hazard regression analysis of PM_2.5_ exposure level associated with the risks of AR and AD. Here, the multivariable Cox model was adjusted for demographic data and comorbidities. The crude HR means non-adjustment for covariates. The adjusted HR means that the adjusted covariates include gender, low income, urbanization level, comorbidity. In the PM_2.5_ exposure level in Q1, patients with AR were at a higher risk for AD (crude HR: 1.454, 95% CI: 0.986–2.142, *p* = 0.0586; adjusted HR: 1.533, 95% CI: 1.023–2.297, *p* = 0.038). In the PM_2.5_ exposure level in Q2–Q4 level, patients with AR were at a significantly higher risk for AD (crude HR: 1.663–8.315; adjusted HR: 1.812–8.981), suggesting that the higher the cumulative exposure to PM_2.5_ for AR patient is, the higher the risk of AD is.

### 3.3. Logistic Regression

The characteristic of logistic regression is the independence of irrelevant alternatives which emphasized the independent contribution of differential cumulative concentration of PM_2.5_ and the AR in AD patients [20,21]. More, we used no matching databases for statistical analysis. Addition, logistic regression the PM_2.5_ exposure at Q1 level as controls in the analysis of case-cohort studies. This trick similar as pervious study [22]. Therefore, the relationship between PM_2.5_ exposure level and risk of AR in AD patients was analyzed by logistic regression better than probit regression, COX regression, and Prentice’s model. Logistic regression analyses were used to calculate the odds ratios (ORs) and CIs for the association between the cumulative level of PM_2.5_ exposure and AR in AD patients. The adjusted OR (CI) for cumulative levels of PM_2.5_ at Q2, Q3, and Q4 were 2.656 (1.666–4.234), 4.805 (2.606–6.401), and 5.604 (3.607–8.706). The results indicate that PM_2.5_ is a risk factor for AR in patients with AD (Figure 3). The results confirm that an increase in PM_2.5_ exposure level increases the risk of AR in AD patients.

## 4. Discussion

Our hypothesis that AR increases the probability of AD, in which exposure to PM_2.5_ increases the higher likelihood of AD, was supported by the present study’s assessment of a national population-based cohort. The major finding of the present study is that Cox proportional hazards analysis, after adjusting for age, sex, low income, and urbanization level, yielded an adjusted HR for AD that was 2.008 times greater for AR than for the comparison cohort. This result accords with other reports that have determined that AR could be a risk factor for AD [23]. One-fifth of patients with AD have allergic diseases, such as bronchial asthma, atopic dermatitis, seasonal AR, perennial AR, and polymorphous atopic disease, and require treatment in Polish rural and urban areas [20]. In addition, 17.7% of patients with AD that have AR require treatment [24]. Expression of proinflammatory mediators, such as interleukin-18, immunoglobulin G, immunoglobulin M, and immunoglobulin E, result in AR in animal and human brain tissue and enhance the pathogenesis of AD [15,25,26]. Numerous disorders that are associated with chronic inflammation are risk factors for AD [27]. In a follow-up study, there was no correlation between AR and hypertension [28]. Comparing the serum of AR patients with healthy patients, it was found that AR changes high-density lipoprotein (HDL) composition and reduces HDL production [29]. This is similar to our results (Table 2), perhaps influenced by AR, leading to a reduced risk of hypertension and hyperlipidemia. Although there is no other research report on the risk of withdrawal of Extrapyramidal signs. Such evidence supports the results of our study, where there was a significantly higher cumulative AD rate in patients with AR.

The present study entailed a systematic review and meta-analysis and observed a strong association between PM_2.5_ exposure and neurological disorders, including AD, stroke, dementia, autism spectrum disorder, and Parkinson’s disease [14]. To date, no studies have reported concentration-dependent effects of PM_2.5_ on the exacerbation of AD. In the present study, there was no correlation between PM_2.5_ exposure level and AD based on a multivariable Poisson regression model (data not shown). However, the concentration-dependent effects of PM_2.5_ on the exacerbation of AR in animal models and human trials have been reported globally [30,31,32]. Therefore, the PM_2.5_ exposure level is a major risk factor for AR [31,33]. Olfactory impairment induced by AR could be caused by sinus chemosensory dysfunction via inflammation in the nasal fossa and around the olfactory neuroepithelium in the olfactory cleft [34]. Olfactory function is the stimulus perception from the intranasal trigeminal nerve endings and the orthonasal and retronasal olfactory airways. Chemosensory activation of the trigeminal nerve is associated with neuroinflammation and neuropathic pain [35]. Patients with AR enhanced the activation of chemosensory trigeminal function and then increased the risk of neuroinflammation [36,37]. On the other hand, the probit regression analyzed the relationship between PM_2.5_ exposure and AR in patients with AD. As shown in Appendix A, the estimated coefficient for cumulative levels of PM_2.5_ at Q2, Q3, and Q4 were 0.327 (0.096–0.557), 0.474 (0.249–0.699), and 0.622 (0.402–0.843), respectively. Similar to the logistic regression model, elevation of the PM_2.5_ exposure level caused the risk of AR. All the aforementioned evidence indicates that PM_2.5_ could be the risk factor for AD by increasing the risk of AR. On the basis of adjusted HRs and ORs in our study, we determined that higher PM_2.5_ exposure levels increase the risk of AR and in turn AD.

The merits of the present study consist of its use of population-based data, which are highly representative of the general population. However, several potential limitations of the present study should be considered. First, the NHIRD does not include data on the behavior and lifestyles of patients; for example, smoking, secondhand smoking, eating habits, and psychological stress. Among these factors, smoking, secondhand smoking, depression, anxiety, and eating habits have been reported to influence the risk for AR and AD. Second, with regard to the PM_2.5_ exposure levels in the patients with AD, data analysis was based on their registered addresses, which were obtained from personal IDs in the NHIRD rather than the actual residences of the patients.

According to the results of the present study, we could purposed that AR is a risk factor for AD, and PM_2.5_ is a risk factor for AR in patients with AD, after the adjustment for PM_2.5_, age, occupational category, urbanization level, and comorbidities. The present study demonstrates that with an increase in PM_2.5_ concentration, the risk of patients with AD having AR will also rise. Our study is the first national population-based cohort study to suggest that PM_2.5_ exposure level is a major risk factor for AR and the further development of AD.

## 5. Conclusions

AD is the irreversible and progressive neurodegenerative and neuroinflammatory disease. AR is an inflammatory and allergic disease of the respiratory system. The Cox proportional hazards analysis purposed that patients with AR had an elevated risk of developing AD. More, AR patients exposed to PM_2.5_ are significantly prone to develop AD. The logistic regression analyses demonstrated that an increase in the PM_2.5_ exposure level at Q2–Q4 increased the risk of AR in AD patients. Our study is the first national population-based cohort study to suggest that PM_2.5_ exposure level is a major risk factor for AR and the further development of AD.

## Figures and Tables

**Figure 1 ijerph-16-03357-f001:**
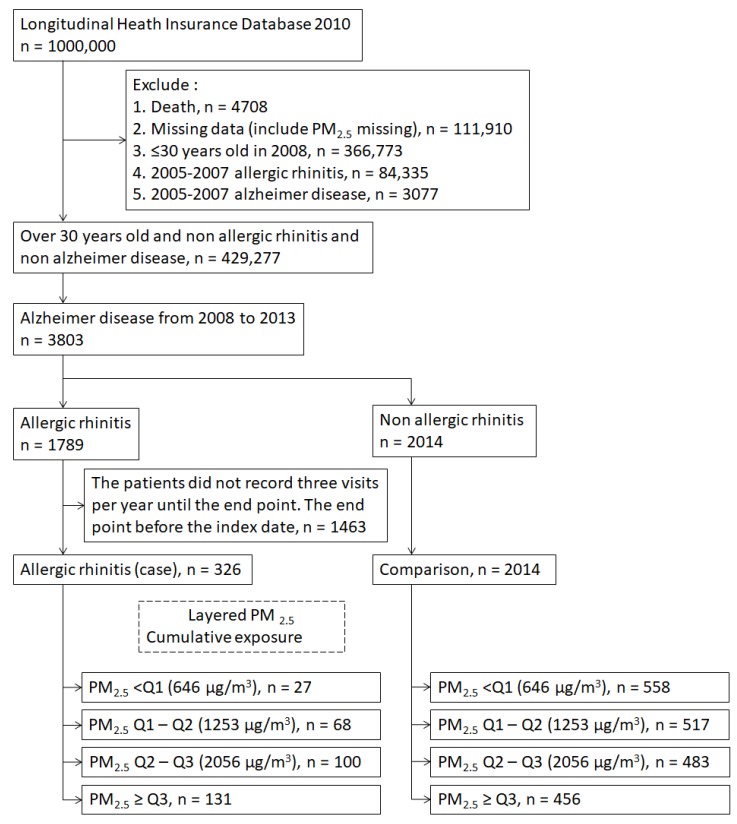
Flow chart of the cohort study.

**Figure 2 ijerph-16-03357-f002:**
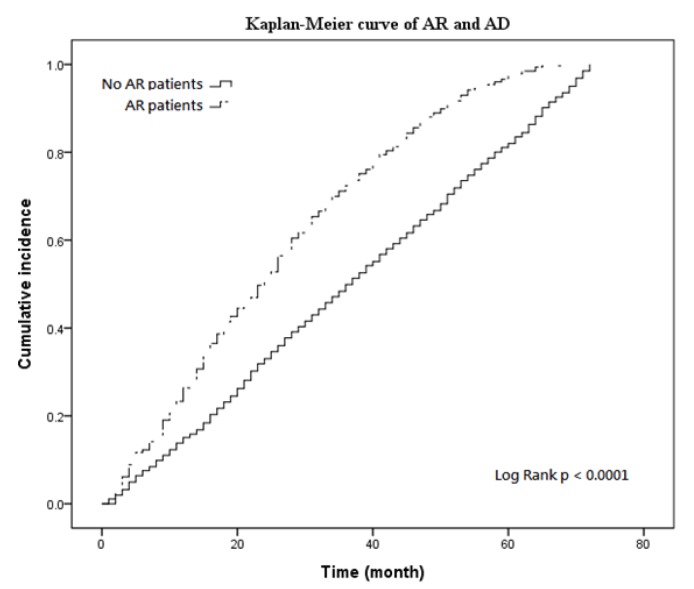
Kaplan–Meier curves estimating cumulative incidence of AD between patients in the AR and comparison cohorts.

**Figure 3 ijerph-16-03357-f003:**
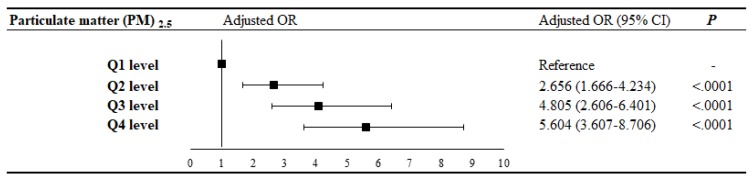
Odds ratio of AR associated with the cumulative PM_2.5_ exposure level.

**Table 1 ijerph-16-03357-t001:** Demographics of individuals with allergenic rhinitis and aged 30 years and above from 2008.

Variable	Comparison		AR		Chi-Square	*p-*Value
	(*n* = 2014)		(*n* = 370)			
Gender						
Female	1088	(54.02%)	158	(48.47%)	3.4787	0.0622
Male	926	(45.98%)	168	(51.53%)		
Age						
30–44	89	(4.42%)	18	(5.52%)	6.0638	0.1085
45–59	241	(11.97%)	50	(15.34%)		
60–74	766	(38.03%)	130	(39.88%)		
>75	918	(45.58%)	128	(39.26%)		
Low income						
Yes	1234	(61.27%)	207	(63.50%)	0.5876	0.4434
No	780	(38.73%)	119	(36.50%)		
Urbanization level						
Highly urbanized	474	(23.54%)	79	(24.23%)	3.5588	0.7818
Moderate urbanization	545	(27.06%)	96	(29.45%)		
Emerging town	291	(14.45%)	44	(13.50%)		
General town	370	(18.37%)	61	(18.71%)		
Aged Township	92	(4.57%)	9	(2.76%)		
Agricultural town	152	(7.55%)	22	(6.75%)		
Remote township	90	(4.47%)	15	(4.60%)		
Particulate matter (PM_2.5_) cumulative exposure						
Q1 level	558	(27.71%)	27	(7.3%)	84.4076	<0.0001
Q2 level	517	(25.67%)	68	(18.38%)		
Q3 level	483	(23.98%)	100	(27.03%)		
Q4 level	456	(22.64%)	131	(35.41%)		

Note: Basic demographic characteristics.

**Table 2 ijerph-16-03357-t002:** The risks of patients with AR versus patients with AD without AR stratified by demographics in Cox proportional hazard regression.

Variable	Alzheimer Disease	
	Adjusted HR (95% CI)	*p-*Value
Allergic rhinitis infection (reference: general population)		
Allergic rhinitis infection	2.008 (1.780–2.266)	<0.0001
Age in 2008 (per year)	1.002 (0.998–1.005)	0.3928
Gender (reference: female)		
Male	0.900 (0.829–0.978)	0.0130
Low income (reference: no)		
Yes	1.129 (1.028–1.240)	0.0113
Urbanization level (reference: Moderate urbanization)		
Highly urbanized	0.968 (0.863–1.085)	0.5759
Emerging town	1.066 (0.933–1.219)	0.3464
General town	0.958 (0.842–1.089)	0.5089
Aged Township	1.051 (0.845–1.309)	0.6537
Agricultural town	0.969 (0.814–1.154)	0.7230
Remote township	1.268 (1.026–1.567)	0.0283
Comorbidity (ref: without)		
Hypertension	0.771 (0.703–0.845)	<0.0001
Hyperlipidemia	0.752 (0.685–0.826)	<0.0001
Diabetes	0.982 (0.894–1.078)	0.6995
Huntington’s Disease	4.024 (0.559–28.99)	0.1670
Parkinson’s Disease	0.980 (0.860–1.116)	0.7566
Extrapyramidal signs	0.773 (0.644–0.926)	0.0053
Sleep Apnea	1.001 (0.656–1.529)	0.9955
Nutrition Deficiencies	0.888 (0.604–1.305)	0.5444

Abbreviation: HR—hazard ratio, CI—confidence interval. Adjusted gender, low income, urbanization level, comorbidity.

**Table 3 ijerph-16-03357-t003:** The risk of patients with AD with AR versus PM_2.5_ exposure level stratified by demographics, crude HR, and adjusted HR in Cox proportional hazard regression.

Cox regression	Particulate Matter (PM) _2.5_ Cumulative Exposure							
	Q1 Level		Q2 Level		Q3 Level		Q4 Level	
	Allergic Rhinitis Infection		Allergic Rhinitis Infection		Allergic Rhinitis Infection		Allergic Rhinitis Infection	
	(Reference: General Population)		(Reference: General Population)		(Reference: General Population)		(Reference: General Population)	
		*p*		*p*		*p*		*p*
Crude	1.454 (0.986–2.142)	0.0586	1.663 (1.290–2.143)	<0.0001	3.128 (2.510–3.898)	<0.0001	8.315 (6.657–10.385)	<0.0001
HR (95% CI)								
Adjusted	1.533 (1.023–2.297)	0.0385	1.812 (1.396–2.353)	<0.0001	3.331 (2.653–4.183)	<0.0001	8.981 (7.100–11.360)	<0.0001
HR (95% CI)								

Abbreviation: HR—hazard ratio, CI—confidence interval. Adjusted gender, low income, urbanization level, comorbidity.

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
