# Peer review of "Influence of PM2.5 Exposure Level on the Association between Alzheimer’s Disease and Allergic Rhinitis: A National Population-Based Cohort Study"

_ijerph, 2019, doi:10.3390/ijerph16183357_

Round 1
Reviewer 1 Report
After revising point by point, I recommend accepting this manuscript.
Author Response
We deeply thank Reviewer #1 for recognition and agreement of our article.
Reviewer 2 Report
The article is well written and the science is sound.
Author Response
We deeply thank Reviewer #2 for recognition and agreement of our article.
Reviewer 3 Report
The author has responded to my comments except the following one point:
authors need to compare Logistic regression with probit regression (or prentice’s model) for their data.
Author Response
We thank the reviewer for suggestions. We have added the compare Logistic regression with probit regression for our data in the section of discussion (on page 11, line 232-236) and supplement table S1.